# Anemia among Men in Malaysia: A Population-Based Survey in 2019

**DOI:** 10.3390/ijerph182010922

**Published:** 2021-10-17

**Authors:** S Maria Awaluddin, Nik Adilah Shahein, Norsyamlina Che Abdul Rahim, Nor Azian Mohd Zaki, Nur Hamizah Nasaruddin, Thamil Arasu Saminathan, Nazirah Alias, Shubash Shander Ganapathy, Noor Ani Ahmad

**Affiliations:** 1Centre for Occupational Health Research, Institute for Public Health, National Institutes of Health, Ministry of Health Malaysia, Setia Alam 40170, Malaysia; 2Centre for Family Health Research, Institute for Public Health, National Institutes of Health, Ministry of Health Malaysia, Setia Alam 40170, Malaysia; drnikadilah@moh.gov.my; 3Centre for Nutrition Epidemiology Research, Institute for Public Health, National Institutes of Health, Ministry of Health Malaysia, Setia Alam 40170, Malaysia; syamlina@moh.gov.my; 4Department of Dietetic & Food Service, Hospital Umum Sarawak, Kuching 93586, Malaysia; adzian_81@yahoo.com; 5Centre for Burden of Disease Research, Institute for Public Health, National Institutes of Health, Ministry of Health Malaysia, Setia Alam 40170, Malaysia; dr_nurhamizah@moh.gov.my (N.H.N.); nazirah.alias@moh.gov.my (N.A.); dr.shubash@moh.gov.my (S.S.G.); 6Centre for Non-Communicable Diseases Research, Institute for Public Health, National Institutes of Health, Ministry of Health Malaysia, Setia Alam 40170, Malaysia; thamilarasu.s@moh.gov.my; 7Director Office, Institute for Public Health, National Institutes of Health, Ministry of Health Malaysia, Setia Alam 40170, Malaysia; drnoorani@moh.gov.my

**Keywords:** anemia, population-based, men, Malaysia, hemoglobin

## Abstract

This study aimed to determine the prevalence of anemia and factors associated with anemia among men in Malaysia. The researchers used data from the 2019 National Health and Morbidity Survey (NHMS). The hemoglobin levels of men aged 15 years and above who gave their consent was measured using the HemoCue^®^ Hb 201+ System©. The majority of them (87.2%) were men aged 15–59 years, referred to as the younger age group in this study. The prevalence of anemia among men was 12.6% (95% confidence interval (CI): 10.9, 14.5). The prevalence was higher among older men (30.7%; 95% CI: 26.6, 35.1) than younger men (10.0%; 95% CI: 8.2, 12.2). Anemia among men was associated with older age (adjusted odds ratios (aOR) = 3.1; 95% CI: 2.1, 4.4) and those with diabetes (aOR = 1.5; 95% CI: 1.2, 2.1) via a logistic regression analysis. In conclusion, older men were more affected by anemia than younger men in this study. Anemia among older men in Malaysia is at the level of moderate to severe public health significance. The likelihood of developing anemia is increased among older men with diabetes compared to older men without diabetes. These often-overlooked issues among men need to be detected and treated early in order to prevent complications and improve their quality of life.

## 1. Introduction

The global prevalence of anemia showed a decreasing trend from 27.0% in 2013 to 22.8% in 2019; however, it still affects nearly one quarter of the world population [1,2]. The prevalence of anemia is described according to the population subgroups, such as men (15–59 years), women of reproductive age (15–49 years), elderly (60 years and above), preschool-aged children (less than five years) and school-aged children (5–14 years) [3]. Although the most affected populations are women and preschool-aged children in developing countries, men are also affected by anemia, especially among the older age group [3]. A local population-based study showed that the prevalence of anemia among men aged 15 years and above was 14.3% in 2015 [4]. A study conducted in India observed a prevalence of 23.2%, while a study conducted in Russia observed a lower prevalence of 5.9% [5,6].

Anemia among men has not been given appropriate attention due to the lower number of affected populations compared to women and preschool-aged children [3,4]. Older men with anemia were grouped with older women because the relationship between older age and chronic diseases is nearly equal in both genders [3]. Anemia among men has usually been discovered accidentally due to other health problems [7,8]. The prevalence of anemia among men increases tremendously among older persons [5]. The prevalence was found to be four times higher among those aged 75 years and above than in younger men [9]. 

Anemia is generally associated with socio-demographic profiles, lifestyle factors and chronic diseases such as chronic kidney disease, diabetes mellitus and cardiovascular disease [10,11,12,13,14]. Anemia among men might also be related to work or lead exposure; however, a substantial proportion of cases are unexplained [15,16,17]. Anemia among men in Malaysia was not given adequate attention, and there is scarce evidence on the factors contributing to the problem. Furthermore, Malaysia’s scheduled national health survey reported an increasing trend of non-communicable diseases’ prevalence, particularly diabetes mellitus [4]. In previous studies, age is suggested as one factor contributing to the likelihood of anemia [9,12]. Thus, a study looking into younger and older men may be beneficial in targeting intervention programs. This study aims to determine the prevalence of anemia according to socio-demographic profiles, lifestyle factors and the presence of chronic illness. The second objective is to determine the associated factors of anemia among men overall, as well as younger and older men in Malaysia.

## 2. Materials and Methods

### 2.1. Study Design and Sampling Procedure

The National Health and Morbidity Survey (NHMS) is a scheduled survey to measure the burden of diseases among the Malaysian population. It used a cross-sectional study design, and national representative respondents were selected via a multistage stratified random cluster sampling. Malaysia was stratified into 13 states and 3 federal territories. Each state was divided into enumeration blocks. An enumeration block (EB) is a geographical mapping of Malaysia according to the number of living quarters (LQs). Each EB contains around 80 to 120 LQs. The average population is approximately 500 to 600 people. The first stage of sampling was conducted by selecting the enumeration block, and the second stage was the selection of living quarters. For each selected living quarter, men aged 15 years and above were eligible for this study. The study’s sample represented the population of men aged 15 years and above in Malaysia, accounting for 11.8 million men.

### 2.2. Data Collection

Socio-demographic data were collected using face-to-face interviews conducted by trained data collectors. Anthropometric measurements such as height and weight were measured using the Tanita Personal Scale HD 319 and the SECA Stadiometer 213. Blood pressure was taken using the Omron Japan Model HEM-907, and capillary blood sampling for point-of-care testing was carried out. The point-of-care testing included the test for fasting blood glucose and cholesterol level via the CardioChek^®^ PA Analyzer [18]. The HemoCue hemoglobinometer (HemoCue^®^ Hb 201+ System, Angelhom, Sweden) was used to check hemoglobin levels. These procedures were conducted by qualified and trained nurses who joined the data collection team. The respondent was appropriately seated and given reassurance prior to the blood pressure examination and the finger prick procedure. The nurses followed the guidelines on safety procedures and clinical waste disposal. The methodology of NHMS 2019 was reported clearly in the technical report and was available on the official website [19]. The sample size for this study was calculated based on the risk factors of diabetes mellitus, with a power of 80%, 95% confidence level and the precision level at 0.05 [13]. The minimum required sample size was 2880 respondents; however, a total of 5079 respondents were included in this study due to the complex survey design and the national representativeness objective.

### 2.3. Ethical Approval and Consent to Participate

This study was registered with the National Medical Research Register (NMRR), bearing registration number NMRR-18-3085-44207. The ethical approval for this study was obtained from the Medical Research and Ethics Committee, Ministry of Health, Malaysia. This study was conducted following the Declaration of Helsinki. Written consent was taken from the respondents before the survey. Additional written consent from parents or legal guardians was documented for respondents aged less than 18 years.

### 2.4. Variables Definitions

The socio-demographic variables included age group, marital status, ethnicity, level of education, place of residence, occupational status and household income. The lifestyle variables were current smoking status, physical activity status and body mass index. Chronic diseases included diabetes mellitus, hypertension and hypercholesterolemia. Anemia in men was defined as hemoglobin levels of less than 13.0 g/dL. It can be further classified into the level severity of mild (11.0–12.9 g/dL), moderate (8.0–10.9 g/dL) and severe anemia (<8.0 g/dL). Hemoglobin levels were adjusted for the respondents who smoked cigarettes by subtracting 0.3 g/dL from the measured level [20]. Body mass index (BMI) was measured according to the Malaysian Clinical Practice Guidelines of Obesity using four categories: underweight (<18.5 kg/m^2^), normal (18.5–22.9 kg/m^2^), overweight (23.0–27.4 kg/m^2^) and obese (<27.5 kg/m^2^) [19]. Respondents known to have diabetes and those with cases newly detected via fasting capillary blood glucose levels of 7.0 mmol/L or higher were defined as having diabetes mellitus. Those with blood cholesterol of more than 5.2 mmol/L were defined as having hypercholesterolemia. Respondents with a blood pressure of ≥140/90 mmHg or known hypertension were considered to have hypertension [21]. The validated short version of the International Physical Activity Questionnaire was used to measure respondents’ physical activity status, categorized into active and inactive [22]. 

### 2.5. Data Analysis

This study used IBM SPSS Statistics for Windows version 21.0 (IBM, Chicago, IL, USA) and R version 4.0.2 software (Microsoft, Redmond, Washington, USA) to generate the findings. The independent variables were tested using simple logistic regression, subsequently followed by multivariable logistic regression. The covariates adjusted for the logistic regression model were age group, marital status, ethnicity, level of education, place of residence, occupational status, household income, current smoking status, physical activity status, body mass index, diabetes mellitus, hypertension and hypercholesterolemia. The results were reported according to the final adjusted model. An odds ratio not equal to one was considered a significant factor. In addition, effect size based on the adjusted odds ratio (aOR) values were considered during data interpretation; small effect (>1.5) and large effect (≥3) [23]. Multicollinearity problems and two-way interaction terms were checked for the final model. The complex sample multivariable logistic regression model fitness was assessed using the complex sample classification table percentage and Akaike Information Criterion (AIC). The complex sample classification table percentage indicates the model predictability and should be more than 70% [24]. The smaller AIC figure in the adjusted model compared to the baseline model indicates that the model fits the data well. The AIC was obtained from R version 4.0.2 software by applying the survey package [25,26]. 

### 2.6. Availability of Data and Materials

The dataset for this study is available; anyone requesting the dataset should send an official data request to the Director General of Health, Ministry of Health, Malaysia.

## 3. Results

The response rate for this study was 94.4% of the total 5079 men, composed of 87.2% (95% CI: 85.8, 88.5) men aged 15–59 years and 12.8% (95% CI: 11.5, 14.2) aged 60 years and above. Among the respondents, 77.5% resided in urban areas, while 75.7% had jobs. Respondents who were not working were either retired persons (15.1%) or students (9.2%). 

The prevalence of anemia was described in three groups: men overall, younger men and older men, versus their socio-demographic profiles, lifestyle factors and chronic diseases. The overall prevalence of anemia among men was 12.6% (95% CI:10.9, 14.5). Further age group categorization showed that the prevalence of anemia among younger men was 10.0% (95% CI: 8.2, 12.2) and 30.7% among older men (95% CI:26.6, 35.1), as shown in Table 1. According to the five-year age group interval, the trend of anemia showed that the rate gradually increases starting with the group aged 55–59 years, as pictured in Figure 1. Table 1 also shows that older men with diabetes have a nearly two times higher prevalence of anemia than older men without diabetes (38.7%; 95% CI: 31.9, 45.9 versus 24.5%; 95% CI: 20.0, 29.7). 

Anemia was associated with older age, place of residence, education level, occupational status, diabetes, hypertension and hypercholesterolemia in the simple logistic regression, as shown in Table 2. In the multivariable logistic regression model, anemia among men was associated with older age (aOR = 3.07; 95% CI: 2.14, 4.41) and those who had diabetes (aOR = 1.53; 95% CI: 1.15, 2.05). The model was adjusted for other socio-demographic factors, lifestyle and other related chronic diseases. A significant interaction term was found involving older age and diabetes (aOR = 1.92; 95% CI: 1.10, 3.37; *p*-value = 0.023). Subsequently, the model was split into two models of older men and younger men. The model among older men indicated that anemia was associated with older men who have diabetes (aOR = 2.45; 95% CI: 1.55, 3.88). However, a similar association was not found in the model of younger men. The model predictability was good, as the complex sample classification table percentage was more than 70% and the AIC was smaller in the full model compared to the baseline model.

## 4. Discussion

Generally, anemia among men may not be given adequate attention by healthcare providers due to a lower prevalence rate and smaller affected populations than women and children [4,6]. This study observed that 12.6% of men in Malaysia had anemia and the prevalence is three times higher among older men. A similar pattern of gradually increasing trends of anemia with older age has been found in previous studies, with the starting point at 50 years and above [9,27]. According to the WHO, anemia’s burden is considered to be of moderate public health significance if the prevalence is between 20 and 39%. A previous local study noted that the trends of moderate public health significance among men started at the age of 50 years, and this study found that it started a bit later, at 60 years [9,20]. This delay indicates some improvement in healthcare services, such as early treatment of anemia among those with chronic illness that might have been carried out [28]. 

The prevalence of anemia among men varies according to age, as discussed above. It also varies according to geographical distribution, ethnicity, household income and diet practice [17,29,30]. The prevalence of anemia in developed countries such as the United States is much lower, at 3.5%, while developing countries such as India have a prevalence of 23.2% [6,31]. A local study recently reported that the prevalence of anemia among men aged 35 to 70 years old was only 4.9% because this study used venous blood sampling and laboratory methods for hemoglobin testing [32]. 

The overall prevalence of anemia among the nationally representative men in Malaysia was 14.3% (96% CI 13.3, 15.4) in 2015 and 12.6% (95% CI: 10.9, 14.5) in 2019 [4,19]. However, the overall prevalence does not capture the burden of anemia in specific groups, such as the very old age group (75 years and above), which has a higher prevalence. Highlighting the anemia problem among the at-risk groups will attract policymakers’ attention to look into the problem seriously. An example of a targeted group for anemia is people with diabetes. A local study found the prevalence of anemia among men with type 2 diabetes and chronic kidney disease was 28.4%, requiring prompt public health attention [28]. Older men who were underweight and who had diabetes were noted to have a higher prevalence of anemia than older men without the mentioned problems. However, the pattern of anemia among younger men was similar across all the socio-demographic factors, lifestyle factors and chronic diseases.

This study also investigated the factors associated with anemia among men. The prevalence rate shows a wide difference between younger and older men. The adjusted model indicated that age and diabetes were associated with anemia among men after controlling for confounding variables. However, age had a significant interaction with diabetes mellitus. Further analysis observed that older men with diabetes were more likely to get anemia than older men without diabetes, while a similar association was not found among younger men in this study. The possible reason for this finding is that older men with diabetes may have diabetic nephropathy or renal anemia [28,33,34]. 

Besides diabetes mellitus, older persons may present with other chronic diseases such as hypertension, ischemic heart disease, cerebrovascular disease and cancer [35]. In addition, older persons with dementia, disabilities and emotional disturbances may impair their food intake and predispose themselves to anemia [36,37]. However, this study only included diabetes mellitus, hypertension and hypercholesterolemia, consistently higher among older men than younger men [4,19]. The association between anemia and hypertension was not found, neither in this study nor a previous study [38]. Hypertension was adjusted in the final model as the disease was prevalent among men in Malaysia and may be related to cardiovascular disease [4,19]. The presence of anemia among those with cardiovascular problems may worsen their health outcomes and quality of life [39,40].

Nonetheless, anemia screening should be conducted regularly among older men as age contributes independently to anemia. Older age alone contributes to anemia, as evidenced by low erythropoietin levels in blood even though the older persons did not have chronic illnesses [41,42]. Hemoglobin levels among healthy older people physiologically decreases with increasing age; however, the cut-off point among older men remained at 13 g/dL [43]. Older persons also tend to have nutrition problems due to lack of food supply, feeding problems, inappropriate diet or malabsorption [29]. 

Among the younger men, the association between diabetes and anemia was not detected due to a small number of younger men who may have had diabetes or developed diabetic nephropathy. This finding also explains that the cause of anemia among younger men might be contributed to by other reasons, such as acute blood loss, genetic disorder or nutritional deficiency. In addition, a longitudinal study noted that younger male adolescents may have nutritional deficiencies even though the prevalence was much lower than younger female adolescents [44].

Further works regarding anemia among men include studying the effectiveness of counseling on diet intake. Moreover, taking adequate nutritional supplementation of iron and healthy food according to food pyramids, in addition to regular medical examination, can be initiated at the individual level. Thus, exploring the barriers to conducting intervention programs among healthcare providers and patients with anemia might be relevant in the future.

### Strengths and Limitations

This study used a survey design, which has an advantage in ensuring Malaysian population-representative respondents. It utilized an online platform via an installed application on tablet devices, where the data could be immediately sent to the central team. It also used validated point-of-care testing and employed well-trained data collectors. Registered nurses were invited to join the data collection, and they were responsible for the clinical procedures. Point-of-care testing has been used widely for population-based screening and has been proven to provide a reliable estimation comparable to the laboratory method [45,46]. The multivariable logistic regression analysis eliminated confounding factors issues compared to univariable, while highlighting interaction issues, which enabled the researcher to interpret findings according to specific groups. This scheduled national population-based survey aimed to measure the burden of non-communicable disease and its risk factors. Hence, much other relevant information regarding anemia was not collected, such as diet history, supplement intake and a specific laboratory measurement of iron profile. Information on the prevalence of chronic kidney disease, peptic ulcer disease, other tropical diseases and lead exposure in the work setting is also necessary in predicting the probability of anemia, which is lacking. The survey used a cross-sectional design and limited the causal relationship between the predictors and the outcome.

## 5. Conclusions

The prevalence of anemia was higher among older men than younger men, and it was associated with older age and diabetes mellitus. Anemia among older men is at the level of moderate to severe public health significance. The likelihood of developing anemia increases among older men with diabetes compared to older men without diabetes. Healthcare providers may overlook anemia among men in Malaysia and only discover it as an incidental finding during hospital admission or primary care settings. Hence, healthcare workers should be aware of anemia problems among older age groups. Regular anemia screening of the targeted group—in this case, older persons—may improve their quality of life and prevent early disease complications.

## Figures and Tables

**Figure 1 ijerph-18-10922-f001:**
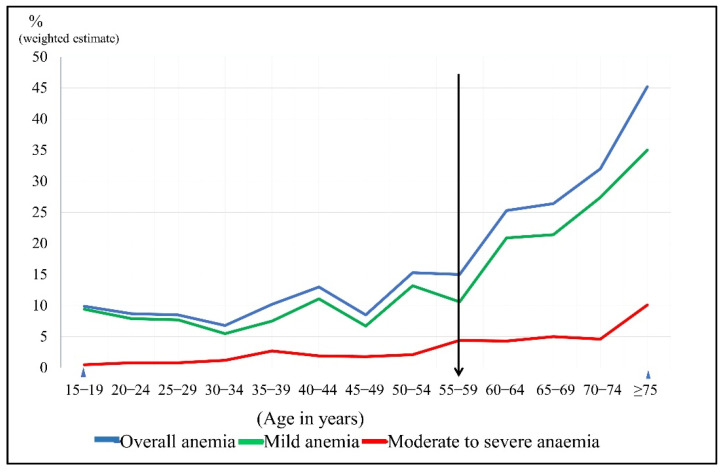
The trends of anemia’s prevalence according to the level of severity using a 5-year age interval among men in Malaysia.

**Table 1 ijerph-18-10922-t001:** Prevalence of anemia among men in Malaysia according to socio-demographic profiles, chronic diseases and lifestyle factors.

Variables	All Men (≥15 Years) *n* = 4793				Younger Men (15–59 Years) *n* = 3738				Older Men (≥60 Years) *n* = 1055			
	*n*	(%)	95% CI		*n*	(%)	95% CI		*n*	(%)	95% CI	
			LL	UL			LL	UL			LL	UL
Overall anemia	732	12.6	10.9	14.5	412	10.0	8.2	12.2	320	30.7	26.6	35.1
Mild anemia	584	10.5	8.9	12.4	325	8.4	6.7	10.6	259	25.0	21.3	29.1
Moderate to severe anemia	148	2.1	1.6	2.6	87	1.5	1.1	2.1	61	5.7	4.0	8.0
Marital status												
Married	546	13.8	12.1	15.7	277	10.2	8.4	12.3	269	29.8	25.4	34.7
Not married	186	10.9	8.4	14.1	135	9.8	7.2	13.2	51	35.9	24.9	48.5
**Strata**												
Urban	395	11.8	9.8	14.2	231	9.3	7.1	12.0	164	30.0	24.9	35.6
Rural	337	15.3	13.0	17.9	181	12.4	9.8	15.6	156	32.7	27.4	38.6
**Ethnicity**												
Malay	479	12.7	11.3	14.3	280	10.3	8.9	12.0	199	29.5	25.0	34.5
Non-Malay	253	12.5	9.6	16.1	132	9.6	6.5	14.0	121	31.8	25.2	39.3
Level of education												
Primary	292	16.2	13.2	19.8	88	9.9	6.9	14.2	204	32.2	27.0	37.9
Secondary	323	11.9	9.7	14.5	239	10.7	8.4	13.6	84	25.8	20.0	32.7
Tertiary	115	10.4	7.9	13.5	84	8.6	6.3	11.8	31	37.5	24.4	52.8
Occupation												
Currently not working	311	18.6	16.0	21.5	84	10.6	8.1	13.7	277	34.3	29.1	40.0
Currently working	420	10.7	8.7	13.1	327	9.9	7.8	12.4	137	24.7	19.1	31.3
Household income group												
Below 40%	340	13.8	11.8	16.2	167	9.6	7.8	11.9	173	33.8	27.9	40.3
Middle 40%	231	11.1	8.9	13.6	156	10.0	7.7	12.8	75	23.8	17.3	31.7
Top 20%	111	12.1	7.8	18.3	80	10.4	6.0	17.3	31	36.3	24.3	50.3
Diabetes												
Yes	277	20.0	17.0	23.5	109	11.6	9.0	15.0	168	38.7	31.9	45.9
No	425	10.9	8.9	13.1	273	9.4	7.4	11.9	152	24.5	20.0	29.7
Hypertension												
Yes	361	17.0	14.7	19.4	129	11.1	8.9	13.7	232	30.1	25.6	35.1
No	341	10.6	8.5	13.2	253	9.3	7.1	12.1	88	32.0	23.7	41.6
Hypercholesterolemia												
Yes	344	15.2	13.1	17.5	157	10.5	8.4	13.0	187	32.1	26.9	37.7
No	358	11.3	9.0	14.0	225	9.4	7.1	12.4	133	29.0	22.8	36.2
Physical activity status												
Inactive	214	14.5	12.1	17.4	88	9.3	6.9	12.4	126	33.6	26.1	42.1
Active	489	12.0	10.0	14.5	304	10.1	7.9	12.9	185	28.3	23.4	33.6
Current smoker												
Yes	238	11.0	8.6	14.1	177	10.0	7.4	13.3	259	33.1	28.3	38.4
No	492	13.8	12.0	15.8	233	10.0	8.2	12.3	61	23.5	17.2	31.2
BMI (Asian cut-off)												
Underweight	51	16.1	10.4	24.0	30	12.1	7.1	19.8	21	44.3	22.1	69.0
Overweight and obese	419	11.6	10.2	13.3	245	9.5	8.0	11.3	174	25.5	20.9	30.6
Normal	180	12.7	8.7	18.3	96	9.7	5.5	16.7	84	34.9	26.7	44.0

All the results were based on weighted estimates.

**Table 2 ijerph-18-10922-t002:** Factors associated with anemia among all men, younger men and older men in Malaysia.

Variables	Crude			Model among Men (≥15 Years)				Model among Younger Men (15–59 Years)				Model among Older Men (≥60 Years)			
	OR	95% CI		aOR#	95% CI		*p*-Value	aOR#	95% CI		*p*-Value	aOR#	95% CI		*p*-Value
		LL	UL		LL	UL			LL	UL			LL	UL	
**Age group (years)**															
15–59	1.00	-	-	1.00	-	-		-	-	-		-	-	-	
≥ 60	3.99	2.92	5.45	3.07	2.14	4.41	<0.001	-	-	-		-	-	-	
**Marital status**															
Married	1.00	-	-	1.00	-	-		1.00	-	-	-	1.00	-	-	
Not married	0.77	0.58	1.02	0.94	0.66	1.33	0.707	0.89	0.61	1.29	0.526	1.40	0.65	3.01	0.384
**Place of residence**															
Urban	1.00	-	-	1.00	-	-		1.00	-	-	-	1.00	-	-	
Rural	1.35	1.02	1.80	1.26	0.95	1.67	0.105	1.33	0.94	1.89	0.106	1.19	0.70	2.00	0.524
**Ethnicity**															
Malay	1.00	-	-	1.00	-	-		1.00	-	-	-	1.00	-	-	
Non-Malay	0.98	0.71	1.35	1.08	0.74	1.58	0.697	1.09	0.65	1.83	0.749	1.05	0.66	1.69	0.824
**Level of education**															
Primary	1.67	1.14	2.45	1.00	0.67	1.52	0.964	0.93	0.52	1.67	0.803	0.86	0.40	1.88	0.711
Secondary	1.17	0.88	1.56	1.25	0.91	1.72	0.174	1.34	0.94	1.92	0.105	0.84	0.39	1.80	0.646
Tertiary	1.00	-	-	1.00	-	-		1.00	-	-	-	1.00	-	-	
**Occupation**															
Currently not working	2.62	1.92	3.56	1.14	0.82	1.57	0.433	0.96	0.59	1.55	0.852	1.32	0.77	2.27	0.309
Currently working	1.00	-	-	1.00	-	-		1.00	-	-	-	1.00	-	-	
**Household income group**															
Below 40%	1.16	0.69	1.97	0.81	0.45	1.47	0.495	0.75	0.37	1.52	0.430	1.02	0.47	2.22	0.961
Middle 40%	0.90	0.57	1.43	0.79	0.49	1.28	0.339	0.84	0.48	1.46	0.534	0.59	0.28	1.25	0.168
Top 20%	1.00	-	-	1.00	-	-		1.00	-	-	-	1.00	-	-	
**Diabetes mellitus**															
Yes	2.06	1.56	2.72	1.53	1.15	2.05	0.004	1.19	0.83	1.70	0.342	2.45	1.55	3.88	0.002
No	1.00	-	-	1.00	-	-		1.00	-	-	-	1.00	-	-	
**Hypertension**															
Yes	1.72	1.30	2.28	1.08	0.80	1.45	0.629	1.17	0.83	1.64	0.365	0.86	0.54	1.38	0.544
No	1.00	-	-	1.00	-	-		1.00	-	-	-	1.00	-	-	
**Hypercholesterolemia**															
Yes	1.41	1.06	1.88	1.12	0.82	1.52	0.467	1.09	0.75	1.58	0.648	1.16	0.73	1.84	0.519
No	1.00	-	-	1.00	-	-		1.00	-	-	-	1.00	-	-	
**Current smoker**															
Yes	0.78	0.59	1.02	0.97	0.71	1.33	0.861	1.02	0.72	1.47	0.894	0.67	0.39	1.15	0.144
No	1.00	-	-	1.00	-	-		1.00	-	-	-	1.00	-	-	
**Physical activity status**															
Inactive	1.24	0.91	1.69	1.06	0.74	1.50	0.755	0.98	0.61	1.59	0.945	1.27	0.76	2.11	0.364
Active	1.00	-	-	1.00	-	-		1.00	-	-	-	1.00	-	-	
**BMI (Asian cut-off)**															
Underweight	1.32	0.68	2.56	1.39	0.69	2.80	0.353	1.31	0.55	3.14	0.542	1.53	0.55	4.30	0.417
Overweight and obese	0.90	0.59	1.39	0.81	0.51	1.30	0.379	0.87	0.46	1.63	0.664	0.63	0.37	1.08	0.092
Normal	1.00	-	-	1.00	-	-		1.00	-	-	-	1.00	-	-	

aOR: adjusted odds ratios; #adjusted for all variables; complex sample classification table: 87.7%; Pseudo R Squares (Nagelkerke: 0.110, Cox and Snell: 0.058, McFadden: 0.079); AIC (full model: 2937.78, baseline model: 3606.4).

## Data Availability

The dataset for this study is available; anyone requesting the dataset should consult the Ministry of Health, Malaysia.

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
