# Peer review of "Anemia among Men in Malaysia: A Population-Based Survey in 2019"

_ijerph, 2021, doi:10.3390/ijerph182010922_

Round 1

Reviewer 1 Report

The topic of this study is interesting for readers in the developing countries.

Generally, the format and contents are good and well-written.

But some questions need to be solved.

1) How about moving the line 134-136 into 2. Materials and methods ?

Then you need to make subtitle for this, ex, study population.

2) The figure of text (line 134, 5079) and the figure of table 1 (4973) are different. What is right? And how did you get the figure of text (line 136, 137, 87.2% 12.8%)? 3788 * 100 / 4973   = 75.2 %

3) Line 158; Table 3->Table 2

Reviewer 2 Report

Thank you for the opportunity to review this interesting manuscript. I have a few minor comments for the authors to consider:

  • Abstract:
    • Line 31: The authors stated that.. The “multivariable” logistic regression “observed” that anemia…. . Since there was only one outcome, i.e. anemia, it was not a “multivariable” logistic regression. The authors simply included covariates in the logistic regression model. Please remove the word “multivariable”. Please also rephrase the sentence (e.g. “It is observed from the logistic regression that anemia…”) because a regression model cannot ‘observe’ anything. Please also rephrase other sentences in the entire manuscript wherever ‘observed’ was incorrectly used.
    • Line 33: The sentence “Older men are more affected by anemia than younger men can be deleted because it is redundant (repetitive of the previously stated results).
  • Introduction:
    • Proofreading and editing of the English language would aid the readability. For instance, “near” needs to be revised to “nearly” in line 42.
    • The authors could provide the definition of anemia (e.g. cut off points of haemoglobin) for each of the age groups (lines 43-45).
    • Lines 51-52: The authors stated that “Anemia among men was not given appropriate attention due to the lower number of affected populations compare to women and preschool-aged children.”. It would be good to clarify if this was the global trend or specific region(s), and include the prevalence of the women and children populations citing the relevant literature.
    • Line 52-54: The sentence “At the same time, older men who had anemia were grouped as anemia among older persons because of the relationship of older age and chronic diseases nearly equal in both genders.” may be deleted because it does not add information to the topic. Alternatively, the authors may revise or rephrase the sentence.
    • Lines 62-63: Whilst the authors stated that “Anemia among men in Malaysia was not given adequate attention and scarce evidence regarding the factor that contributes to the problem”, it would be good to also highlight the need to investigate the research gap. For instance, information about the relatively high prevalence of chronic kidney disease (CKD), diabetes mellitus (DM) and cardiovascular disease (CVD) in Malaysia. It would be good to also justify the reason of studying the impact by age groups by relating to the relevant latest literature.
  • Methods:
    • Lines 71-72: The authors stated that “Each state was divided into enumeration blocks”. Please clarify the criteria used to divide the blocks. For instance, was it the socioeconomic status or % of the population, etc.
    • Line 74: Please add if the questionnaire used to collect socio-demographic data was assessed for validity (e.g. face or content or construct validity). If none of these was not assessed, the authors could add this as a limitation of the present study in the discussion section.
    • Line 82: Please add a sentence to clarify if the trained nurses (and trained data collectors noted on line 75) have good inter-rater reliability, and how this was assessed. If the reliability was not assessed, the authors could add this as a limitation of the present study in the discussion section.
    • Line 89: The authors stated that 2,880 respondents was the required sample size. Please clarify if this calculation was based on 80% or 90% power, and 5% significance? Please also justify the choice of using diabetes mellitus (not CVD or CKD) for the calculation.
    • Line 116: Please add the definition used to categorise physical activity status based on the IPAQ.
    • Line 119: Please delete “The complex samples analysis was utilized” because it was not complex analyses. The new sentence (lines 118-119) would read “This study used IBM SPSS Statistics for Windows version 21.0 and also R version 118 4.0.2 software to generate descriptive, bivariate and logistic regression analysis”.
    • Please delete “multivariable” on line 120 and line 121 because there was only one outcome, (i.e. anemia), and not multiple outcome variables. It is incorrect to use “multivariable” in this case.
    • Line 122: Please add the list of the covariates adjusted for in the logistic regression model in a new sentence.
    • Line 125-126: Please specify the interaction terms used in the model, was it all combination, or just a few, e.g. age*diabetes mellitus, smoking*CVD, etc
    • Line 132: Please add the contact details, e.g. email address for the readers to contact MoH.
  • Results:
    • Line 134: Please add a new sentence to clarify if the 5,079 men were in proportion to the overall states and territories or were they mainly from a few states. In other words, were they representative of the entire Malaysian population? The authors could also comment on the representativeness of their sample in terms of the other socio-demographic variables.
    • Table 2: Please add a footnote to clarify the covariates used in the adjusted logistic regression model. Please also clarify what the asterisk (*) implies.
  • Discussion:
    • Lines 195-196: The authors stated that the prevalence was “more representative”. Please clarify what the authors compared it to, e.g. more representative than the other studies?
    • Line 219: Please clarify if the higher prevalence in older men was observed in the present study or from the cited literature.
    • The authors could discuss the findings against the literature published in the region, e.g. Asia or South East Asia or even the ASEAN countries, to enhance the depth of the discussion.
    • Line 244: Please comment on the reliability scores of the non lab-based measurements (e.g. the socio-demographic data).
    • Line 245: Please delete the word “multivariable” because it is incorrect.
    • The authors may add specific recommendations for future studies and/or highlight the impact of the research findings.
  • Conclusions:
    • Lines 257-259, lines 260-261: Please remove two sentences: “Taking adequate nutritional supplementation of iron and healthy food according to food pyramids besides regular medical examination can be initiated at the individual level.”, and “Regular anemia screening of the targeted group may improve quality of life and prevent early disease complications.” because the authors did not study the impact of doing so on anaemia or QoL or complications in the present study. Please do not overpromise. The authors could suggest these as directions for future studies in the Discussion section instead.

Reviewer 3 Report

This manuscript presented an interesting topic that investigates the often overlooked factors associated with the prevalence of anemia in Malaysia, using data from a national survey.  The paper is well-written with clear statement of the problem, data, results and conclusions, etc. I only have one major comment regarding the details of the data analysis section.

Major comment:

  1. Line 118, the authors mention that the study used IBM SPSS and R for data analysis. I am wondering which part of the analysis used SPSS/R. For R, did the author use any R package for analysis? If the answer is yes, please specify the name of the R package.
  2. Line 125, the authors mention that multicollinearity problems were checked. I am wondering how they checked multicollinearity. Please add 1-2 sentence in the data analysis section and 1-2 sentence in the result section explaining why multicollinearity is not a problem in their analysis.
  3. Line 125, the authors mention that two-way interaction terms were checked for the final model. Did the author check each pair of the variables in the final model? In that case, would multiple comparison be a problem?
  4. Line 126-127 the authors mention that the complex sample multivariable logistics regression model was assessed using the complex sample classification table percentage and AIC. Please add 1-2 sentence explaining the details and also 1-2 sentence in the result section.

Minor comments:

  1. In Table 1, 2, and Figure 1, I would recommend adding that all the results were weighted so that the readers do not need to go back to the data analysis section to find the information.
  2. In Figure 1, points could be added for better presentations of the data. The legends for the lines should be longer.
  3. In Table 2, what does "aOR#" mean? Is it a typo?

Author Response

Please see the attachment at the subtopic Response to Reviewer 3 Comments.

Reviewer 4 Report

Overall, I think that this is a well conducted study concerning a topic which is likely to be of interest to readers of the journal. The data has also been nicely analysed and interpreted and the draft manuscript is generally well written. That said, I have a few minor comments which the authors may wish to consider:

  • R software has been used to perform some of the data analysis. Were any specific packages used? If so which ones were they?
  • In line 158 you mentioned Table 3. Do you mean Table 2?
  • In Table 2 please make the work ‘hypercholesterolema’ bold to match the others.
  • Can line 185-186 be reworded please as this doesn’t make any sense to me.
  • In line 206 change ‘factor’ to ‘factors’ please as you looked at more than one.
  • Line 212 needs some citations to support its statement.
  • You mention in lines 234-235 that the anemia in younger men may be caused by nutritional deficiencies. Is there any evidence of this in the Malaysian population? Some more discussion around this point would be helpful.

Author Response

Please see the attachment at the subtopic Response to Reviewer 4 Comments.
